# A Hydrodistillate of *Gynostemma pentaphyllum* and Damulin B Prevent Cisplatin-Induced Nephrotoxicity In Vitro and In Vivo via Regulation of AMPKα1 Transcription

**DOI:** 10.3390/nu14234997

**Published:** 2022-11-24

**Authors:** Minhyeok Song, Minseok Kim, Dang Hieu Hoang, Lochana Mangesh Kovale, Jihyun Lee, Youngjoo Kim, Changhyun Lee, Jongki Hong, Sungchul Park, Wonchae Choe, Insug Kang, Sung Soo Kim, Joohun Ha

**Affiliations:** 1Department of Biochemistry and Molecular Biology, Graduate School, College of Medicine, Kyung Hee University, Seoul 02447, Republic of Korea; thdalsgur77@naver.com (M.S.); mskim9262@naver.com (M.K.); hoang.dang.hieu.ls@gmail.com (D.H.H.); kovlelochana@gmail.com (L.M.K.); wchoe@khu.ac.kr (W.C.); iskang@khu.ac.kr (I.K.); sgskim@khu.ac.kr (S.S.K.); 2Easy Hydrogen Corporation, Jeju City 63196, Republic of Korea; easyhydrogen@gmail.com; 3Department of Urology, College of Medicine, Jeju National University, Jeju City 63243, Republic of Korea; kurology@jejunu.ac.kr; 4Chunjieh Cooperation, Jeju City 63359, Republic of Korea; 7771rkstlr@hanmail.net; 5College of Pharmacy, Kyung Hee University, Seoul 02447, Republic of Korea; jhong@khu.ac.kr (J.H.); yrs01004@naver.com (S.P.)

**Keywords:** *Gynostemma pentaphyllum*, damulin B, AMPKα1, nephrotoxicity, cisplatin

## Abstract

The clinical application of cisplatin, one of the most effective chemotherapeutic agents used to treat various cancers, has been limited by the risk of adverse effects, notably nephrotoxicity. Despite intensive research for decades, there are no effective approaches for alleviating cisplatin nephrotoxicity. This study aimed to investigate the protective effects and potential mechanisms of a Gynostemma pentaphyllum leaves hydrodistillate (GPHD) and its major component, damulin B, against cisplatin-induced nephrotoxicity in vitro and in vivo. A hydro-distillation method can extract large amounts of components within a short period of time using non-toxic, environmentally friendly solvent. We found that the levels of AMP-activated protein kinase α1 (AMPKα1), reactive oxygen species (ROS), and apoptosis were tightly associated with each other in HEK293 cells treated with cisplatin. We demonstrated that AMPKα1 acted as an anti-oxidant factor and that ROS generated by cisplatin suppressed the expression of AMPKα1 at the transcriptional level, thereby resulting in induction of apoptosis. Treatment with GPHD or damulin B effectively prevented cisplatin-induced apoptosis of HEK293 cells and cisplatin-induced acute kidney injury in mice by suppressing oxidative stress and maintaining AMPKα1 levels. Therefore, our study suggests that GPHD and damulin B may serve as prospective adjuvant agents against cisplatin-induced nephrotoxicity.

## 1. Introduction

Cisplatin, one of the most potent chemotherapy agents available, is widely used for the treatment of a variety of solid tumors, but more widespread use of this agent is limited because of its severe adverse effects, including nephrotoxicity, gastrointestinal toxicity, neurotoxicity and hepatotoxicity [1]. In particular, AKI is a major side effect of cisplatin treatment, with current statistics indicating that clinical nephrotoxicity occurs in about one-third of cancer patients treated with cisplatin [1,2]. AKI is a syndrome characterized by a rapid, potentially reversible decline in renal function, including a rapid fall in glomerular filtration rate and acute tubular cell death, over a period of hours and days [3]. Multifactorial processes, including DNA damage, generation of ROS, disruption of mitochondrial function, induction of inflammation, activation of cellular apoptosis and necrosis, have been implicated in the development of cisplatin-induced AKI [2]. Although diverse pharmacological and molecular strategies for reducing the side effects of cisplatin have been developed, their clinical applications are still limited owing to the highly complex etiology of AKI [2,4]. In this context, natural products and herbal medicines harboring multiple biological activities are currently attracting considerable research attention for their potential to alleviate cisplatin-induced AKI [5].

Gynostemma pentaphyllum, a perennial liana widely cultivated in Southeast Asian countries, including Korea, Southern China, Japan and India, has been widely used as medicinal herb, herbal tea, and dietary supplement owing to its notable pharmacological activities [6]. Among these properties are anti-cancer, anti-diabetic, anti-obesity, anti-inflammation, cardioprotective, and neuroprotective effects [7]. The multiple pharmacological effects of this plant are attributable to its various components, including saponins, flavonoids, and organic acids [7]. In particular, dammarane-type saponins, termed gypenosides, have been the subject of intensive research interest owing to their numerous beneficial effects; to date, 328 dammarane-type saponins have been isolated from Gynostemma species and structurally defined [8]. Toxicological studies have shown that extracts of *G. pentaphyllum* are relatively safe in both acute and long-term toxicity experiments [8]. Despite multiple reports showing various beneficial effects of extracts of this plant, there is limited information on the potential protective effects of extracts or any of its components against cisplatin-induced AKI.

AMPK is composed of a catalytic subunit (α) and two regulatory subunits (β and γ), each with multiple isoforms [9]. AMPKα1 is ubiquitously expressed in all tissues, whereas AMPKα2 shows a tissue-specific expression pattern, with high expression in metabolically active tissues, including muscle, heart, and liver [10]. Although it has been well established that AMPK plays a central role in regulating cellular and whole-body energy homeostasis [9], a role of AMPK under DNA-damaging conditions, such as cisplatin treatment, is highly controversial. It was reported that AMPK contributes to cisplatin-induced apoptosis in A549 cells [11] and mediates the effect of metformin in enhancing the chemosensitivity of hepatoma cells to cisplatin [12]. On the other hand, some studies have reported that AMPK mediates the protective effects of ginsenoside against cisplatin-induced cytotoxicity [13] or that AMPK activation protects normal tissues against cisplatin-induced toxicities [14].

In the present study, we report the protective effects of GPHD and its major component, damulin B, a dammarane-type saponin, against cisplatin-induced cytotoxicity and AKI in mice. We further examined the mechanisms underlying these protective effects in vitro and in vivo, demonstrating the involvement of transcriptional regulation of AMPKα1 and ROS generation.

## 2. Materials and Methods

### 2.1. Reagents and Antibodies

Dulbecco’s modified Eagle’s medium (DMEM) powder was purchased from Gibco (Grand Island, NY, USA). Antibodies against Bax, Bad, cleaved PARP-1, Bcl-2, β-actin, and HA-Tag, and small interfering RNA (siRNA) against AMPKα1 were purchased from Santa Cruz Biotechnology (Santa Cruz, CA, USA). Antibodies against AMPKα1, cleaved caspase-7, cleaved caspase-3, phospho-ATR (Ser428), and phospho-histone H2A.X (Ser139) were purchased from Cell Signaling Technology (Danvers, MA, USA). The DNA transfection reagent and siRNA transfection reagents were purchased from Thermo Fisher (Waltham, MA, USA). Damulin B was obtained from ChemFaces Biochemical (Wuhan, China).

### 2.2. Preparation of Cell Culture Media Containing a Hydrodistillate of G. pentaphyllum Leaves

The hydrodistillate of *G. pentaphyllum* leaves was provided by Chunjieh Cooperation (Jeju City, Korea). For large-scale production, 20 kg of dried *G. pentaphyllum* leaves were mixed with 1.2 tons of distilled water in a high pressure steam tank and heated to 109 °C at a pressure of 1.5 atmospheres for 15 h. During the heating step, vapor was condensed by cooling in a closed-loop system. The collected hydrodistillate was further sterilized by heating at 95 °C for 1 h. DMEM powder was directly dissolved in 100% (*v/v*) hydrodistillate, and the pH was adjusted to 7.2 with bicarbonate. After filter sterilization, fetal bovine serum and antibiotics (Corning, NY, USA) were added to 10% and 1%, respectively. The medium was further mixed with prepared DMEM to obtain the desired percentage (*v/v*).

### 2.3. Animal Experiments

Three-week-old male ICR mice were purchased from Japan SLC (Shizuoka, Japan) and allowed to adapt for 1 wk at 21 ± 2 °C and 50% ± 5% relative humidity under a 12-h light/12 h dark cycle before use in experiments. Experimental mice (5 mice/group) were fed a normal diet and GPHD (~0.9 mL/day) for 28 d, whereas the negative control group was fed a normal diet and an equal amount of distilled water. Thereafter, mice were intraperitoneally injected with cisplatin (20 mg/kg) or dimethyl sulfoxide (vehicle). Four days later, blood samples and kidney tissues were collected for analysis. In another set of experiments, 4-wk-old mice (5 mice/group) were intraperitoneally injected with damulin B (25 or 50 mg/kg) or vehicle every day for 7 d and fed a normal diet and distilled water. Mice were then intraperitoneally injected with cisplatin (20 mg/kg) or vehicle, and blood samples and kidney tissues were collected for analysis 4 d later. The animal protocol was approved by the Institutional Animal Care and Use Committee of Kyung Hee University (KHSASP-21-309).

### 2.4. Assessment of Blood Parameters and Histopathological Analysis

Mouse blood was collected and centrifuged to obtain plasma. Plasma was mixed with reagents for analyzing BUN and CRE, purchased from SEKISUI Medical (Tokyo, Japan), and then analyzed using an AU480 automated analyzer (Beckman Coulter, California, USA). NGAL was analyzed using an enzyme-linked immunoassay kit (Abcam, Cambridge, UK). For histopathological analyses, mouse kidneys were immediately dissected out, fixed in 10% neutral buffered formalin solution, embedded in paraffin, and cut at a thickness of 5 µm. Each section was subjected to H&E and PAS (BBC Biochemical, WA, USA) staining and colorimetric TUNEL assay (Merck Millipore, Billerica, MA, USA), carried out. Subsequent histopathological examinations were performed using a light microscope (Olympus).

### 2.5. Primers Used for Reverse Transcriptase-PCR

Human AMPKα1, 5′-GTGGCTCACCCAACTATGCT-3′ (forward) and 5′-CCCTTG GTGTTTCAGCAACC-3′ (reverse); human β-actin, 5′-ACGTTGCTATCCAGG CTGTG-3′ (forward) and 5′-AAGGAAGGCTGGAAGAGTGC-3′ (reverse); mouse AMPKα1, 5′-AGCCGACTT TGGTCTTTCAAAC-3′ (forward) and 5′-GTGGTCGTCC AGGAAAGAGT-3′ (reverse); and mouse β-actin, 5′- ATCTACGAGGGCTATG -3′ (forward) and 5′-GGAAGGCTGGAAAAGAGCCT-3′ (reverse). The level of each mRNA was normalized to that of β-actin.

### 2.6. Miscellaneous Experiments

Western blot analysis, apoptosis assay, reporter gene assay, cell viability assay, statistical analysis [15], ROS measurement [16] were performed as previously described.

## 3. Results

### 3.1. GPHD Protects HEK293 Cells from Cisplatin-INDUCED Cytotoxicity

To extract bioactive compounds from *G. pentaphyllum* leaves, we used a hydro-distillation method that employs a non-toxic, environmentally friendly solvent and is capable of extracting large amounts of components within a short period of time. To examine the therapeutic potential of *G. pentaphyllum* extracts, we prepared cell culture media containing the *G. pentaphyllum* hydrodistillate (hereafter, GPHD), as described in Materials and Methods. Cisplatin induced cytotoxicity of HEK293 cell in a concentration-dependent manner, reducing cell viability by ~40% following treatment at a concentration of 20 μM for 24 h, whereas GPHD alone showed no cytotoxicity at a concentration up to 50~75% (*v/v*) (Figure 1A). Therefore, we used 20 μM cisplatin and 50% GPHD in subsequent studies. Under these conditions, pretreatment with GPHD significantly alleviated cisplatin-induced cytotoxicity (Figure 1B). The protective effect of GPHD against cisplatin-induced cytotoxicity was further demonstrated using a fluorescence-based analysis of 7-AAD and annexin V double-positive cells (Figure 1C) and Western blot analysis of apoptotic markers, including Bad, Bax, cleaved caspase-3, caspase-7, and PARP-1, and the DNA damage markers, phospho-ATR and phospho-H2AX (Figure 1D).

### 3.2. Cisplatin Decreases the Expression of AMPKα1 at the Transcriptional Level

Although it has been well established that AMPK plays a central role in the regulation of carbohydrate and lipid metabolism, its role under DNA-damaging conditions is highly controversial. We thus examined whether AMPK is involved in regulating the fate of HEK293 cells in the setting of cisplatin treatment. Cisplatin decreased the expression of AMPKα1 in a concentration-dependent manner (Figure 2A), an effect that occurred at the transcriptional level, as evidenced by a decrease in AMPKα1 mRNA levels and its promoter activity (Figure 2A,B). Notably, both phenomena were effectively blocked by GPHD (Figure 2C,D). These data indicate that AMPKα1 is regulated at the transcriptional level in response to cisplatin and its level is inversely correlated with the degree of cytotoxicity.

### 3.3. GPHD Protects HEK293 Cells from Cisplatin-Induced Cytotoxicity through AMPKα1

We next attempted to clearly elucidate the role of AMPKα1 in the context of cisplatin and GPHD treatment. Overexpression of AMPKα1 in HEK293 cells significantly attenuated cisplatin-induced apoptosis and potentiated the preventive effect of GPHD (Figure 3A,B). Conversely, siRNA-mediated knockdown of AMPKα1 sensitized cells to cisplatin-induced apoptosis and diminished the protective effect of GPHD (Figure 3C,D). Collectively, these results indicate that AMPKα1 plays a critical role in cell survival, and that GPHD effectively alleviates cisplatin-induced cytotoxicity by maintaining AMPKα1 levels in the face of inhibitory effects of cisplatin.

### 3.4. GPHD Suppresses Cisplatin-Generated Oxidative Stress via AMPKα1

Cisplatin induces cytotoxicity through multiple signal pathways, one of the most important of which is generation of oxidative stress. Cisplatin treatment dramatically increased cellular ROS levels in HEK293 cells (Figure 4A), and treatment with the strong anti-oxidant N-acetyl cysteine (NAC) effectively decreased cisplatin-induced ROS generation (Figure 4A) and apoptosis (Figure 4B,C). Moreover, NAC concomitantly prevented cisplatin-induced suppression of AMPKα1 expression and the promoter activity (Figure 4C,D). These data suggest that cisplatin inhibits the transcription of AMPKα1 and induces apoptosis in an ROS-dependent manner. GPHD treatment effectively blocked cisplatin-induced ROS generation, highlighting the anti-oxidant properties of GPHD (Figure 4E). Overexpression of AMPKα1 significantly blocked cisplatin-induced ROS generation and further potentiated the anti-oxidant effects of GPHD (Figure 4E). Conversely, knockdown of AMPKα1 promoted cisplatin-induced ROS generation and inhibited the anti-oxidant effects of GPHD (Figure 4F). Collectively, these data indicate that AMPKα1 plays a critical role as an anti-oxidant cellular factor and its expression is inhibited by ROS, revealing a complex signaling loop between ROS generation and AMPKα1 regulation.

### 3.5. Intake of GPHD Prevents Cisplatin-Induced AKI in Mice

We further examined the effect of GPHD on cisplatin-induced AKI. To this end, 4-wk-old mice were fed GPHD or distilled water and a normal diet and for 28 days. Each group consumed essentially equal volumes of distilled water or GPHD (~0.9 mL/d). Cisplatin (20 mg/kg) was then intraperitoneally injected, and 4 d later, mice were sacrificed for analysis (Figure 5A). GPHD intake did not affect body weight, but effectively prevented the weight loss induced by cisplatin (Figure 5B). Moreover, GPHD intake significantly attenuated cisplatin-induced tubular damage, as indicated by serum levels of BUN, creatinine, and NGAL (Figure 5C). A histological examination of mouse kidneys by H&E staining, PAS staining, and TUNEL assay revealed that GPHD intake significantly prevented cisplatin-induced tubular necrosis, cast formation, and apoptosis (Figure 5D). A Western blot analysis of kidney tissue extracts revealed that GPHD significantly reduced cisplatin-induced apoptosis and DNA-damage (Figure 5E). In accord with results obtained in HKE293 cells (Figure 2), GPHD intake also rescued the cisplatin-induced decrease in AMPKα1 mRNA and protein levels in mouse kidneys (Figure 5E).

### 3.6. Damulin B Prevents Cisplatin-Induced Cytotoxicity in HEK293 Cells and Cisplatin-Induced AKI in Mice

To analyze active components, we lyophilized GPHD and dissolved the powder in methanol for further analysis by ultra-high performance liquid chromatography-quadrupole/time of flight mass spectrometry (UHPLC-Q/TOF MS) in negative ion mode (Figure 6).

Several types of gypenosides, hydroxylated dammarane-type glycosides, and damulins were detected as major components and tentatively identified based on their LC elution order and exact mass measurements. We further investigated the effect of the single compound, damulin B, which is the most abundant component of GPHD, and two commercially available gypenosides, XLVI and XVII. These gypenosides share a dammarane-type scaffold characteristic of tetracyclic triterpenoids (Figure 7A). Among the three tested compounds—none of which showed cytotoxicity towards HEK293 cells at a concentration of 30 μM (Figure 7A)—damulin B was the most efficacious in preventing cisplatin-induced cytotoxicity (Figure 7B). Similar to the case for GPHD, damulin B effectively blocked cisplatin-induced apoptosis (Figure 7C,D) and ROS generation (Figure 7E) in HEK293 cells and rescued cisplatin-induced suppression of AMPKα1 mRNA and protein expression (Figure 7D) and AMPKα1 promoter activity (Figure 7F). The effect of damulin B was further examined in vivo using a cisplatin-induced AKI mouse model. An analysis of serum parameters (Figure 8A), a histological examination (Figure 8B), and Western blot analysis of kidney lysates (Figure 8C) revealed that damulin B attenuated cisplatin-induced AKI. It also restored mRNA and protein levels of AMPKα1 in mouse kidneys (Figure 8C).

## 4. Discussion

To date, 328 different gypenosides have been identified from various Gynostemma species, exhibiting a wide range of pharmacological activities both in vitro and in vivo [8]. However, their potential against cisplatin-induced nephrotoxicity and renal injury has received little research attention, and their underlying pharmacological and molecular mechanisms are essentially unknown. In the present study, we demonstrate for the first time that a hydrodistillate of *G. pentaphyllum* leaves (GPHD) and its major component, damulin B, are effective agents against cisplatin-induced renal injury in vitro and in vivo, and further report the novel findings that that AMPKα1 mediates the protective effects of GPHD and damulin B against cisplatin-induced renal injury.

AMPK exhibits complex allosteric regulation by adenosine nucleotides (e.g., AMP, ADP, and ATP), reflecting different cellular energy levels. In addition, phosphorylation by upstream kinases such as LKB1 and CAMKK2 fully activates AMPK [9,17]. Therefore, the major regulatory mechanisms for AMPK occur at the protein level, resulting in rapid changes in enzymatic activity. Thus, it is a highly intriguing observation that AMPKα1 is regulated at the transcriptional level by cisplatin, increasing our understanding of the complexity of AMPK regulatory mechanisms. Specifically, we found that cisplatin inhibited the transcription of AMPKα1, and that this phenomenon was effectively attenuated by GPHD and damulin B (Figure 2, Figure 5, Figure 7 and Figure 8). Although the precise mechanisms by which cisplatin suppresses AMPKα1 transcription remains unknown, our data suggest that ROS is an important factor because treatment with the general antioxidant NAC significantly blocked cisplatin-induced ROS generation and simultaneously attenuated the inhibitory effect of cisplatin on AMPKα1 transcription (Figure 4). Our data further imply a highly complex signaling relationship between AMPKα1 and ROS generation. In accord with a previous report showing that AMPK is involved in the ROS defense system [16], our data demonstrated that AMPKα1 exerts an anti-oxidant effect in the context of cisplatin treatment. Specifically, we showed that overexpression of AMPKα1 significantly decreased cisplatin-induced ROS generation whereas AMPKα1 knockdown increased it (Figure 4). Thus, our data suggest that the balance between AMPKα1 and ROS levels contributes to determining the fate of cisplatin-treated cells. Another process that may contribute to the protective effects GPHD and damulin B is autophagy. According to a recent report, cisplatin-induced AKI is associated with the induction of autophagy, which acts as a cytoprotective mechanism for cell survival [18]; notable in this context, AMPK is also known as a positive regulator of autophagy [19]. Collectively, these observations suggest that a strategy that maintains AMPKα1 level could be further developed as an adjuvant therapy to protect kidney cells from cisplatin-induced injury and thereby reduce its adverse effect.

In sharp contrast to the well-established role of AMPK in metabolic regulation [9]. there is no consensus regarding whether AMPK is activated or inhibited by genotoxic anticancer agents, including cisplatin and doxorubicin. Moreover, it is not clear whether AMPK acts as a survival factor or contributes to cell death in the context of cisplatin treatment. Contributing to this uncertainty, different isoforms of AMPKα are likely to have opposing effects in response to genotoxic agents. As demonstrated in the current study, activation of AMPKα1 or maintenance of AMPKα1 levels can protect cells against cisplatin-induced toxicity. Notably, we previously reported that AMPKα2 exerts proapoptotic effects in doxorubicin-treated H9c2 cardiomyocytes by interfering with mitochondrial integrity and function and that doxorubicin induced transcription of AMPKα2 in H9c2 cells [20]. In contrast, doxorubicin inhibited the transcription of AMPKα1 in several cancer cell lines [15], observations similar to those in the current study. Thus, genotoxic agents are likely to simultaneously inhibit expression of the survival factor AMPKα1 and induce expression of the proapoptotic factor AMPKα2. Therefore, the balance between the level of two isoforms, AMPKα1 and AMPKα2, may determine cellular susceptibility to genotoxic agents, including cisplatin. Discrepancies observed in previous reports regarding the activity, level, or role of AMPK in the setting of cisplatin or doxorubicin treatment may reflect the complex differential regulation and roles of AMPKα isoforms.

The numerous pathological conditions responsible for the development of AKI include sepsis, interstitial nephritis, trauma, renal ischemia/reperfusion, administration of toxins, and side effects of anticancer drugs such as cisplatin and doxorubicin [21]. An increasing number of studies have shown that AKI and chronic kidney disease (CKD) are closely linked, and likely promote one another, and are therefore currently considered as an integrated clinical syndrome [22]. Although AKI and CKD are affecting millions of people worldwide and causing high mortality, no specific treatment capable of attenuating these worldwide health problems has yet emerged [22]. To understand the underlying mechanisms or develop therapeutic interventions, researchers have developed numerous animal models that simulate clinical conditions of renal failure, reflecting the multiple causes of renal failure [23,24]. Of these, the cisplatin-induced AKI model is known to be highly reproducible and similar to human renal disease [23]. Although the etiology of AKI is diverse, generation of ROS appears to be a common phenomenon during various types of AKI [25]. Both GPHD and damulin B show very strong anti-oxidant effects, as demonstrated here. Thus, it will be interesting to see whether future studies show that these agents exert similar protective effects in other AKI models in addition to the cisplatin-induced model.

## 5. Conclusions

In the present study, we demonstrated the potential of GPHD and its major component, damulin B, as protective agents against cisplatin-induced nephrotoxicity and AKI, suggesting that GPHD and damulin B may serve as prospective adjuvant agents against cisplatin-induced nephrotoxicity.

## Figures and Tables

**Figure 1 nutrients-14-04997-f001:**
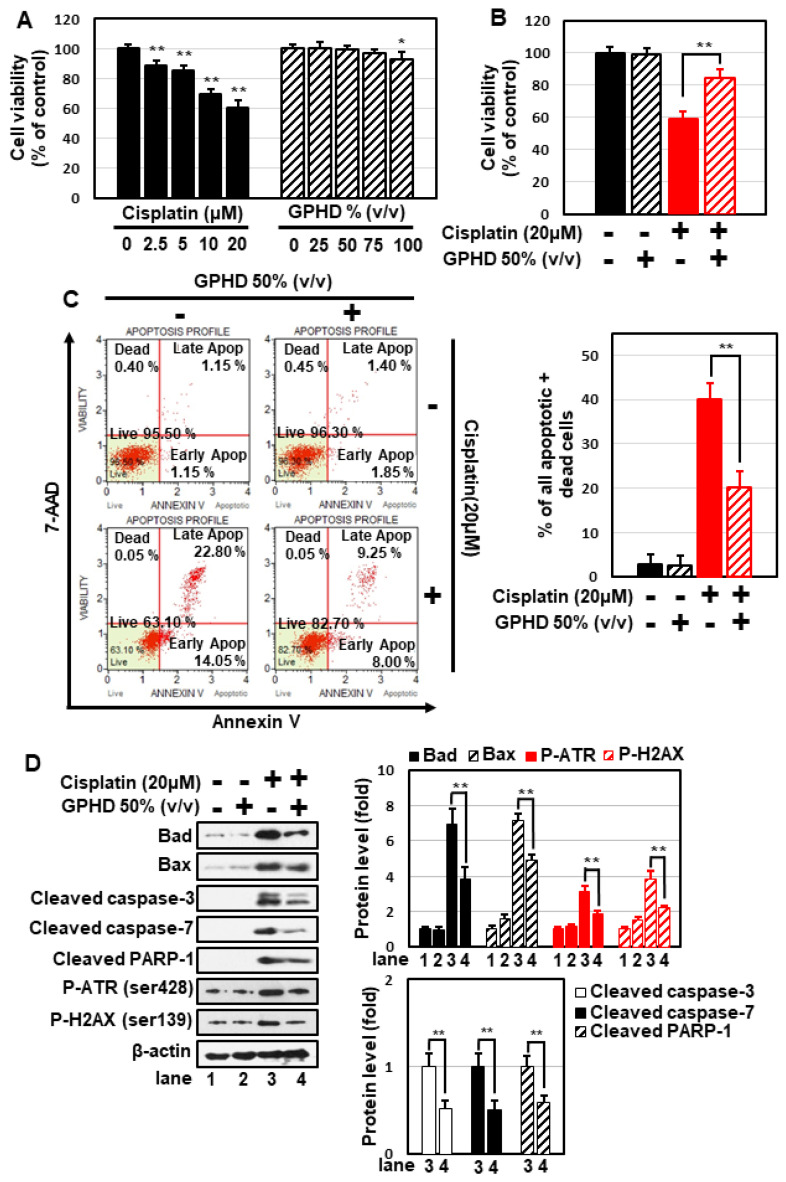
Preventive effects of GPHD on cisplatin-induced cytotoxicity in HEK293 cells. HEK293 cells were treated with the indicated concentration of cisplatin and/or GPHD for 24 h, then cell viability assays (**A**,**B**), FACS analysis of 7-AAD and annexin V double-positive stained cells (**C**), and Western blotting (**D**) were performed. The results shown are representative of the three independent experiments. *p*-value < 0.05 was considered statistically significant; individual *p*-values (* *p* < 0.05; ** *p* < 0.01) are indicated in figures.

**Figure 2 nutrients-14-04997-f002:**
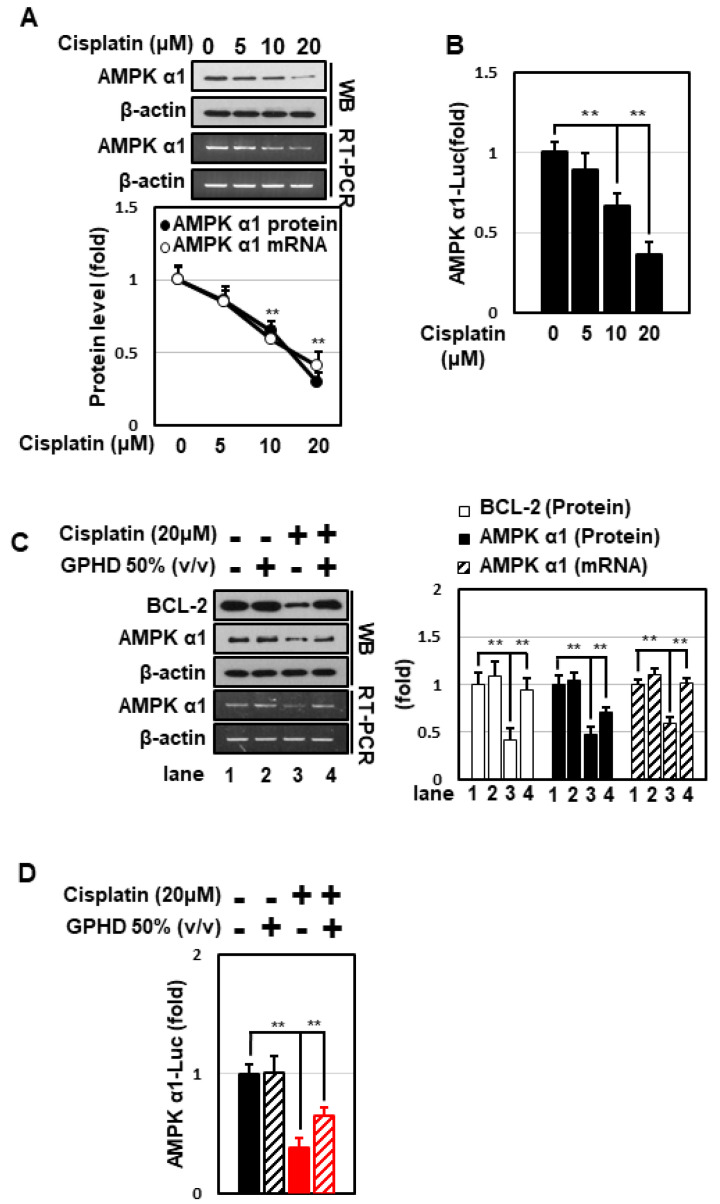
GPHD effectively prevents cisplatin-induced inhibition of AMPKα1 transcription. HEK293 cells were treated with cisplatin and/or GPHD for 24 h, and then Western blot analysis or RT-PCR was performed (**A**,**C**). HEK293 cells were transfected with a pGL3 luciferase reporter vector containing a human AMPKα1 promoter (~1.7 kb). After the indicated treatment for 24 h, luciferase activity was measured (**B**,**D**). WB, Western blotting; RT-PCR, reverse transcriptase-polymerase chain reaction. The results shown are representative of the three independent experiments. *p*-value < 0.05 was considered statistically significant; individual *p*-values (** *p* < 0.01) are indicated in figures.

**Figure 3 nutrients-14-04997-f003:**
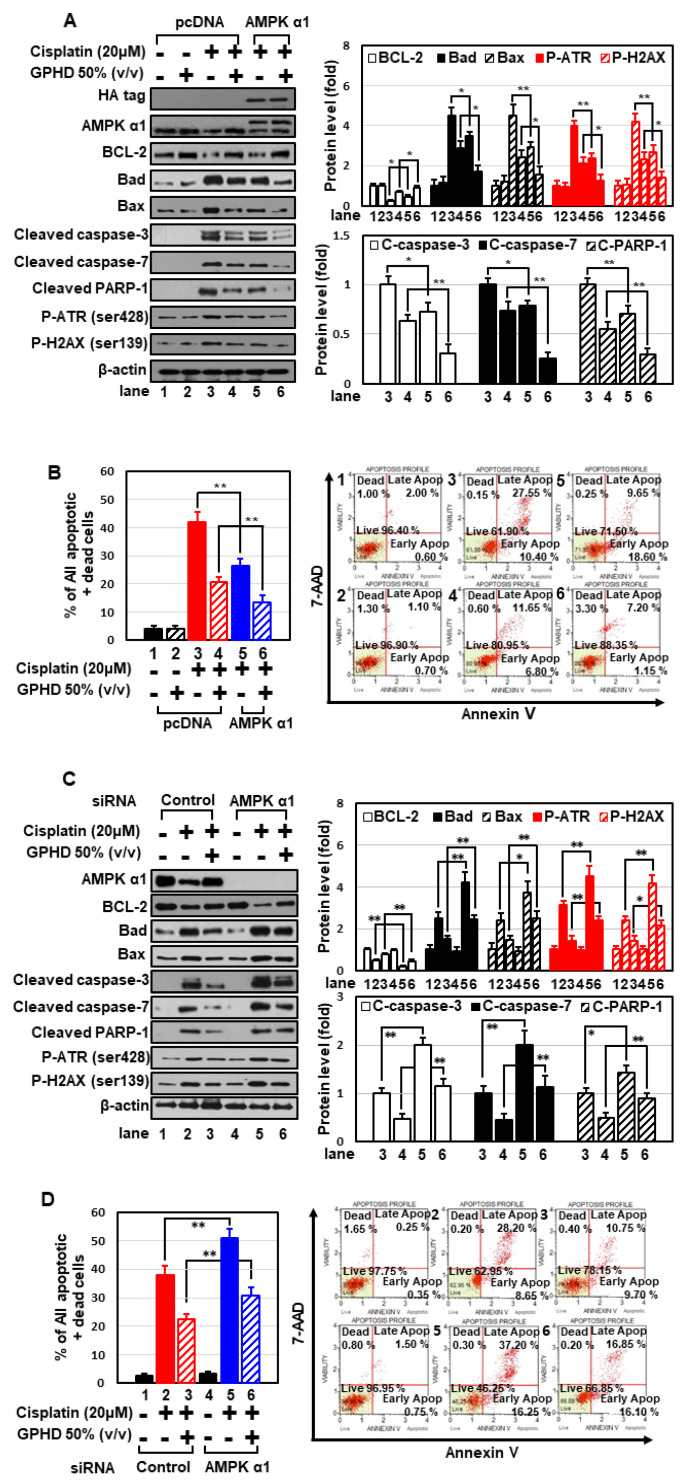
AMPKα1 plays a critical role in cell survival in the context of cisplatin and GPHD treatment. HEK293 cells were transfected with an expression vector for HA-tagged AMPKα1 (**A**,**B**) or siRNA for AMPKα1 (**C**,**D**), and then were treated with cisplatin and/or GPHD for 24 h. Apoptosis was analyzed by Western blotting (**A**,**C**) and FACS analysis (**B**,**D**). The results shown are representative of the three independent experiments. *p*-value < 0.05 was considered statistically significant; individual *p*-values (* *p* < 0.05; ** *p* < 0.01) are indicated in figures.

**Figure 4 nutrients-14-04997-f004:**
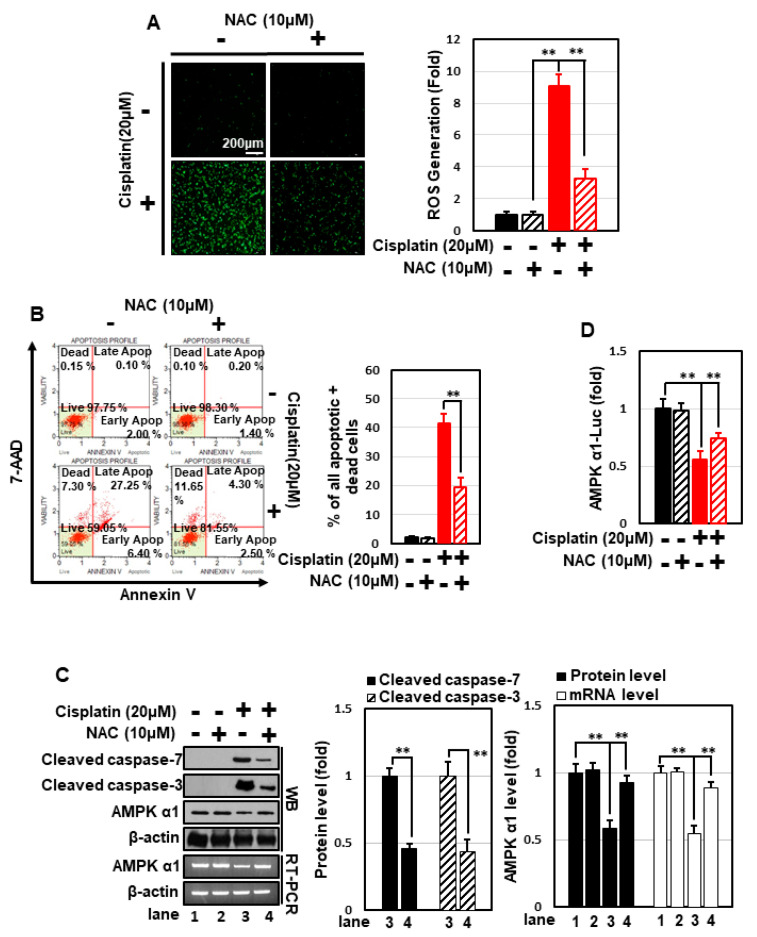
GPHD suppresses cisplatin-generated ROS and ROS-induced apoptosis via AMPKα1, a cellular anti-oxidant factor. HEK293 cells were treated with cisplatin (20 μM) and/or NAC (10 μM) for 24 h. Thereafter, intracellular ROS levels were measured (**A**), and FACS analysis of 7-AAD and annexin V double-positive cells (**B**), Western blot analysis (**C**), and luciferase reporter assays for AMPKα1 promoter activity (**D**), were performed. HEK293 cells were transfected with an expression vector for HA-tagged AMPKα1 (**E**) or siRNA for AMPKα1 (**F**), then treated with cisplatin and/or GPHD for 24 h, after which cellular ROS levels were measured. The results shown are representative of the three independent experiments. *p*-value < 0.05 was considered statistically significant; individual *p*-values (** *p* < 0.01) are indicated in figures.

**Figure 5 nutrients-14-04997-f005:**
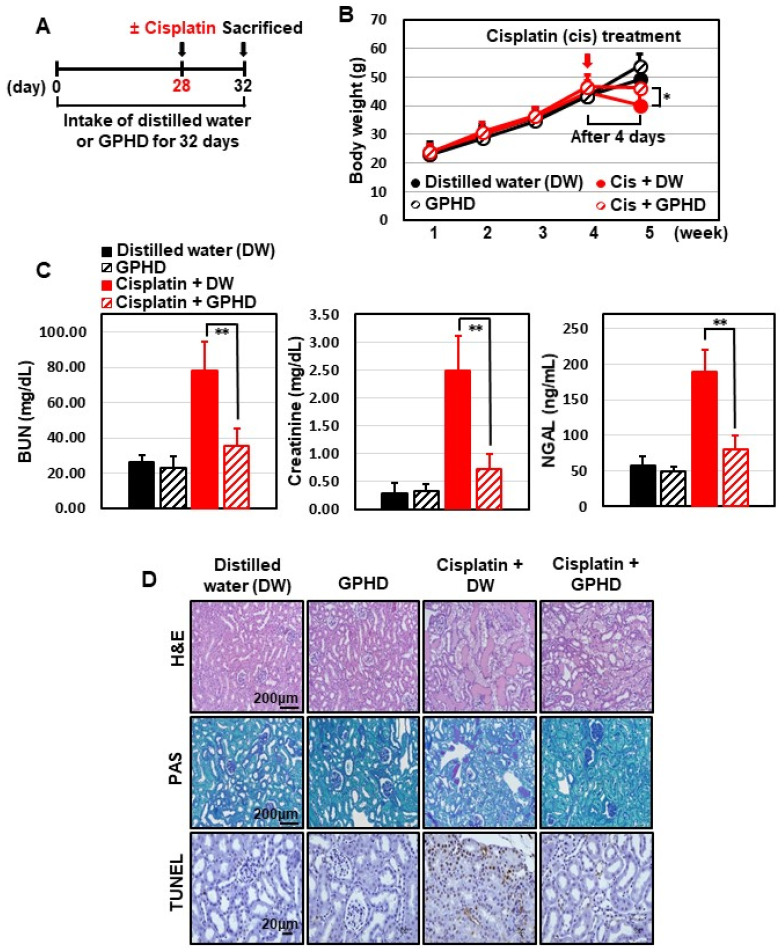
Effects of GPHD on cisplatin-induced AKI in mice. Four-week-old mice were fed a normal diet and GPDH or distilled water (~0.9 mL/d) for 28 days prior to intraperitoneal injection of cisplatin (20 mg/kg). Four days after cisplatin injection, mice were sacrificed (**A**). Mice were weighed (**B**), blood urea nitrogen (BUN), creatinine, neutrophil gelatinase-associated lipocalin (NGAL) in serum were measured (**C**), tissues were histologically examined by H&E, PAS, and TUNEL staining (**D**), and kidney tissue extracts were analyzed by Western blotting (**E**). *p*-value < 0.05 was considered statistically significant; individual *p*-values (* *p* < 0.05; ** *p* < 0.01) are indicated in figures.

**Figure 6 nutrients-14-04997-f006:**
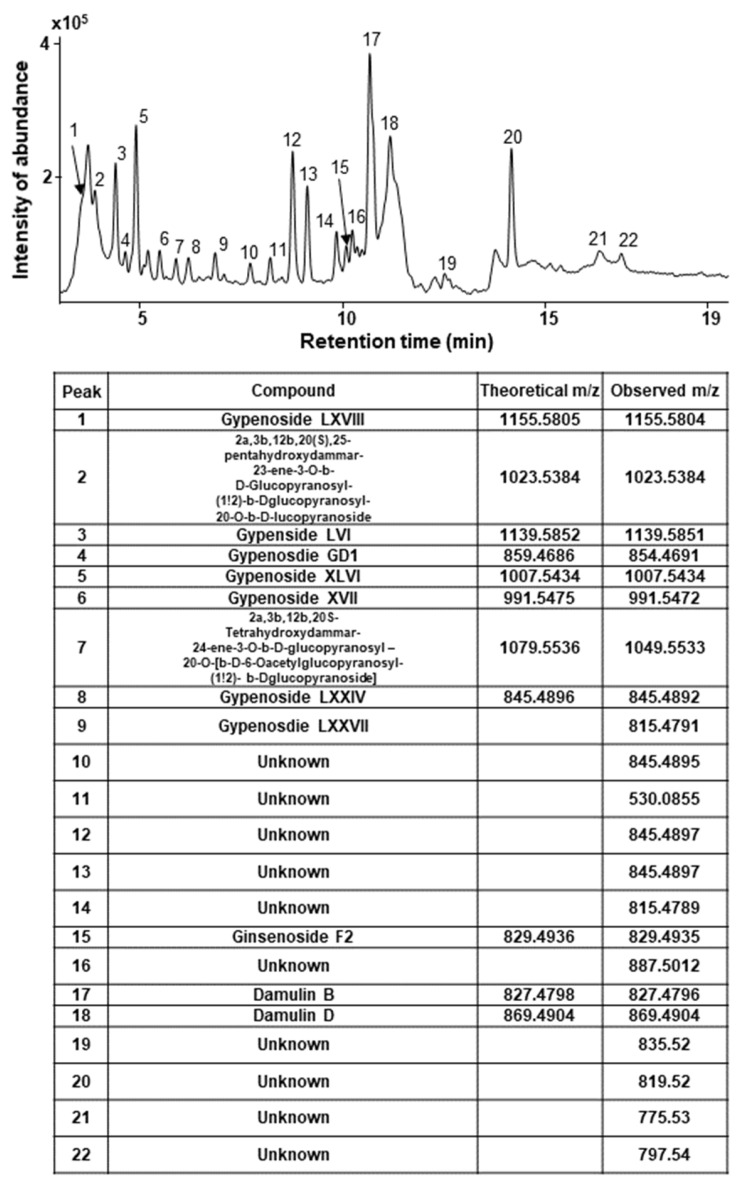
LC-MS spectrum of GPHD. GPHD was analyzed by UHPLC-Q/TOF MS in negative ion mode. Several types of gypenosides, hydroxylated dammarane-type glycosides, and damulins were detected as major components and tentatively identified based on their LC elution order and exact mass measurements.

**Figure 7 nutrients-14-04997-f007:**
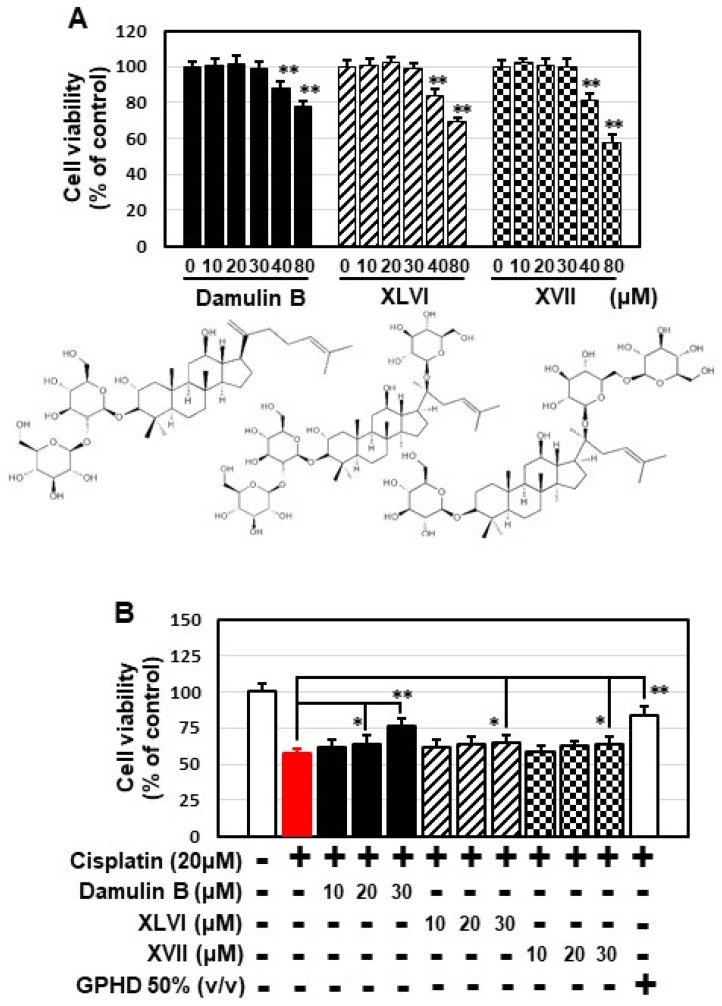
Damulin B protects HEK293 cells from cisplatin-induced cytotoxicity. HEK293 cells were treated with the indicated concentration of damulin B or gypenoside XLVI or XVII in the presence or absence of cisplatin for 24 h, and then analyzed. (**A**,**B**) Cell viability assay; (**C**) apoptosis assay; (**D**) Western blotting and RT-PCR analysis; (**E**), cellular ROS level; (**F**) luciferase assay. The results shown are representative of the three independent experiments. *p*-value < 0.05 was considered statistically significant; individual *p*-values (* *p* < 0.05; ** *p* < 0.01) are indicated in figures.

**Figure 8 nutrients-14-04997-f008:**
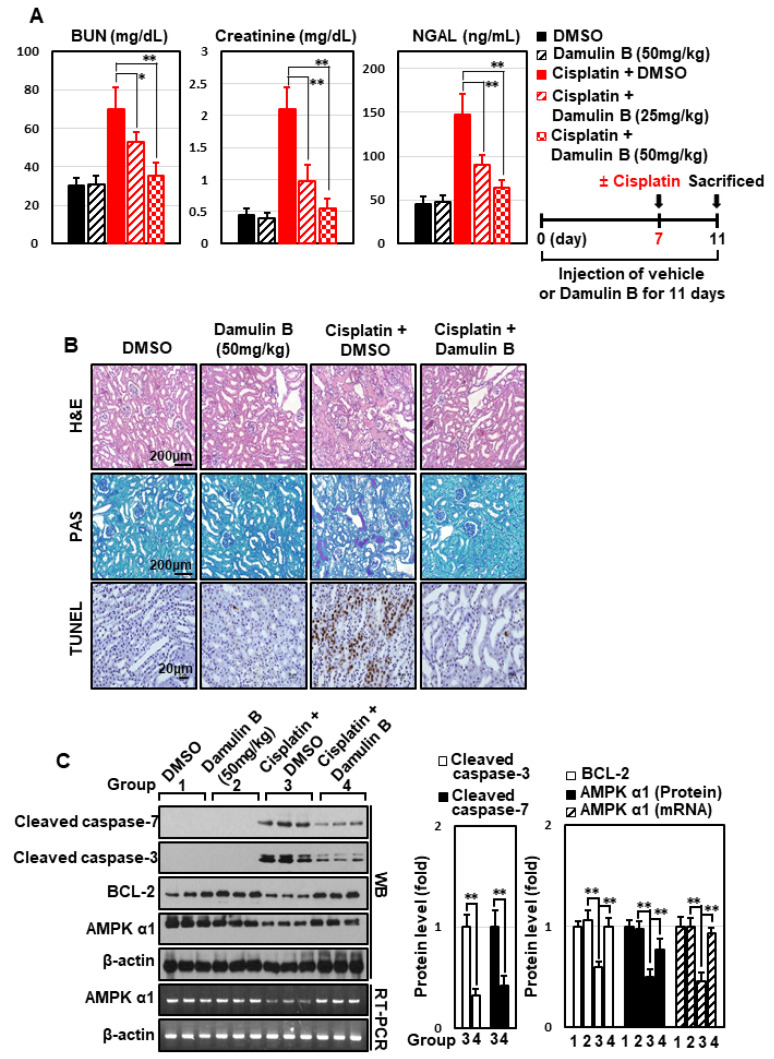
Damulin B prevents cisplatin-induced AKI in mice. Mice were intraperitoneally injected with damulin B (25 or 50 mg/kg) or vehicle every day prior to intraperitoneal injection of cisplatin (20 mg/kg). Four days after cisplatin injection, animals were sacrificed. (**A**) Serum BUN, creatinine, and NGAL levels; (**B**) histological examination of kidney tissue via H&E, PAS, and TUNEL staining; (**C**) Western blot analysis of kidney extracts. *p*-value < 0.05 was considered statistically significant; individual *p*-values (* *p* < 0.05; ** *p* < 0.01) are indicated in figures.

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
