# Peer review of "A Hydrodistillate of Gynostemma pentaphyllum and Damulin B Prevent Cisplatin-Induced Nephrotoxicity In Vitro and In Vivo via Regulation of AMPKα1 Transcription"

_nutrients, 2022, doi:10.3390/nu14234997_

Round 1

Reviewer 1 Report

This study investigated the protective effects and potential mechanisms of a Gynostemma pentaphyllum leaves hydrodistillate (GPHD) and its major component, damulin B, against cisplatin-induced nephrotoxicity. It was shown that the treatment with GPHD or damulin B effectively prevented cisplatin-induced apoptosis of HEK293 cells and cisplatin-induced acute kidney injury in mice by suppressing oxidative stress and maintaining AMPKα1 levels. The study has accordingly proposed that GPHD and damulin B may serve as prospective adjuvant agents against cisplatin-induced nephrotoxicity.The study is reasonably designed and well conducted and the data is well presented, which altogether make the work the acceptable quality. 

Author Response

We thank you for your kind evaluation of the current manuscript.

Reviewer 2 Report

The presented data are interesting, however I have some comments to methodology. Materials and methods   Was the weight of the mice controlled during the experiment? From what data were selected the compounds concentration used in in vivo studies? In what material were parameters to Western blot analysis determined?

Author Response

We thank you for your kind evaluation on the current manuscript.

The presented data are interesting, however I have some comments to methodology. Materials and methods  

  1. Was the weight of the mice controlled during the experiment?

Reply: We initially purchased 3-week-old male ICR mice with an essentially identical weight for in vivo experiments, as explained in page 3 of the Material section, and we indeed monitored the weight of the mice under each experimental condition and the results were summarized and indicated in figure 5A, page 11.

  1. From what data were selected the compounds concentration used in in vivo studies?

Reply: In this study, to extract bioactive compounds from G. pentaphyllum leaves, we used a hydro-distillation method that employs water, a non-toxic, environmentally friendly solvent and is capable of extracting large amounts of components within a short period of time. Accordingly, we found that G. pentaphyllum hydrodistillate did not show cytotoxicity at a concentration of 75%, and very slight cytotoxicity at 100% in HEK293 cells (Fig. 1A). We observed that each mouse consume ~0.9 ml per day. For these collective reasons, the mice of experimental conditions were fed equal amount of G. pentaphyllum hydrodistillate, ~0.9 ml per day. This procedure is explained at page 3, animal experiment section, and at the legend of figure 5.

3.In what material were parameters to Western blot analysis determined?

Reply: The antibody used for each Western blot was indicated at the left side of each panel, and beta-actin was used for the loading control. The intensity of each band was measured and quantified based on that of beta-actin.